# The Impact of Microbiota on the Pathogenesis of Amyotrophic Lateral Sclerosis and the Possible Benefits of Polyphenols. An Overview

**DOI:** 10.3390/metabo11020120

**Published:** 2021-02-20

**Authors:** Julia Casani-Cubel, María Benlloch, Claudia Emmanuela Sanchis-Sanchis, Raquel Marin, Jose María Lajara-Romance, Jose Enrique de la Rubia Orti

**Affiliations:** 1Doctoral Degree School, Catholic University of Valencia San Vicente Mártir, 46001 Valencia, Spain; juliacasani@mail.ucv.es; 2Department of Health Science, Catholic University San Vicente Mártir, 46001 Valencia, Spain; joseenrique.delarubi@ucv.es; 3Laboratory of Cellular Neurobiology, School of Medicine, Faculty of Health Sciences, University of La Laguna, 38190 Tenerife, Spain; rmarin@ull.edu.es; 4Multimedia Department, Catholic University of Valencia San Vicente Mártir, 46001 Valencia, Spain; jlajara@ucv.es

**Keywords:** amyotrophic lateral sclerosis, polyphenols, microbiota

## Abstract

The relationship between gut microbiota and neurodegenerative diseases is becoming clearer. Among said diseases amyotrophic lateral sclerosis (ALS) stands out due to its severity and, as with other chronic pathologies that cause neurodegeneration, gut microbiota could play a fundamental role in its pathogenesis. Therefore, polyphenols could be a therapeutic alternative due to their anti-inflammatory action and probiotic effect. Thus, the objective of our narrative review was to identify those bacteria that could have connection with the mentioned disease (ALS) and to analyze the benefits produced by administering polyphenols. Therefore, an extensive search was carried out selecting the most relevant articles published between 2005 and 2020 on the PubMed and EBSCO database on research carried out on cell, animal and human models of the disease. Thereby, after selecting, analyzing and debating the main articles on this topic, the bacteria related to the pathogenesis of ALS have been identified, among which we can positively highlight the presence mainly of *Akkermansia muciniphila*, but also *Lactobacillus* spp., *Bifidobacterium* spp. or *Butyrivibrio fibrisolvens*. Nevertheless, the presence of *Escherichia coli* or *Ruminococcus torques* stand out negatively for the disease. In addition, most of these bacteria are associated with molecular changes also linked to the pathogenesis of ALS. However, once the main polyphenols related to improvements in any of these three ALS models were assessed, many of them show positive results that could improve the prognosis of the disease. Nonetheless, epigallocatechin gallate (EGCG), curcumin and resveratrol are the polyphenols considered to show the most promising results as a therapeutic alternative for ALS through changes in microbiota.

## 1. Introduction

Neurodegenerative diseases involve several disorders that are characterized by a progressive loss of neurons in different regions of the central nervous system and the brain. Currently, the most prevalent diseases are possibly Alzheimer’s disease (where the neurons that die are the pyramidal neurons of the hippocampus) and Parkinson’s disease (where the neurons of the substantia nigra are affected). Motor neuron diseases (MND) include a wide and heterogeneous group of clinical entities that are characterized by progressive degeneration, especially motor neurons. The most common clinical phenotype is amyotrophic lateral sclerosis (ALS), described for the first time by Jean-Martin Charcot in 1869, which can be classified as familial ALS with a genetic component and representing 10% of cases [1], or sporadic ALS, which represents 90% of the remaining cases and does not show any conventional hereditary pattern [2]. The majority of patients are aged between 50 and 75 years when diagnosed and, despite the disease currently being classified as a rare disease, it is estimated that approximately 400,000 patients will be diagnosed with ALS in the whole world by 2040 [3]. ALS is outlined due to its severity with life expectancy, which is between three and five years in the majority of patients. This is due to the quick progressive death of motor neurons. Depending on the kind of motor neuron that is initially altered, the patient can be diagnosed with bulbar onset ALS when affecting first-order motor neurons and spinal onset ALS when the neurons that are affected are second-order motor neurons [4]. Motor neurons go from the brain to the spinal cord and muscles throughout the body and they establish the necessary communication between the brain and the voluntary muscles. Their deterioration and gradual death lead to a loss in muscular functioning, and voluntary and involuntary muscle paralysis [5]. As a result, patients show progressive difficulty in speaking, swallowing and finally breathing, resulting in death [6]. This neuronal death is etiologically related to excess glutamate and mitochondrial alterations that cause neurodegeneration [7,8,9]. Glutamate is the main neurotransmitter of the central nervous system (CNS) and is essential for neuronal synapses, yet it is toxic for neurons when found in high concentrations [10]. High levels of glutamate have been observed in the serum of ALS patients [11], which is positively correlated with the severity of the disease [12]. Nonetheless, the role of the γ-aminobutyric acid (GABA) neurotransmitter also seems relevant for the etiopathogenesis of the disease, as the receptor block mainly induces temporary muscle hyperexcitability and motor deficits that are associated with moderate loss of motor neurons [13]. Therefore, the balance between the excitatory activity of glutamate and the inhibitory activity of GABA largely depends on the viability of the motor neuron. Regarding energetic alterations, the mitochondrion is a vital organelle of the cell with multiple functions. It is the main source of ATP, maintaining calcium homeostasis and taking part in its signaling. Thus, malfunctioning mitochondria affect neurons, especially motor neurons [14,15]. In this sense, in the alteration of mitochondrial energy balance, as evidenced in ALS patients, the alteration has an important role in the metabolism of tryptophan based on catabolism and activation of the kynurenine pathway. Said pathway is the de novo NAD+ synthesis associated with cell energy, repairing and fatigue, which explains that the dysregulation of this metabolism is related to the risk of developing neurodegenerative diseases [16], and especially in ALS [17,18]. Thus, it seems relevant to have adequate levels of tryptophan, a precursor of serotonin, as according to several authors enhancing the serotonergic pathway is a therapeutic alternative for the disease. In SOD1 rodents (G86R), a transgenic model of ALS, serotonin levels are lowered in the brain stem and spinal cord before motor symptoms appear. In addition, there is an important atrophy of serotonin-containing neuronal cell bodies alongside neuritic degeneration at the onset of the disease [19]. Furthermore, serotonergic denervation leads to a loss of control on the inhibition of glutamate release, thus increasing its levels in the synaptic cleft, and identifying the importance of the serotonergic system at a neuroprotective level [20]. On the other hand, there has been evidence that there is a link between neurotoxicity due to high levels of heavy metals and the disease [21].

In relation to these neurotransmitters associated with the disease, certain strains of bacteria play an essential role as they can modulate the production of GABA and glutamate [22] or serotonin [23]. This gives us an idea of how important human microbiota is for brain activity, as it is comprised of billions of bacteria that reside on or inside the body [24].

The human body, including the intestine, skin and other mucous membranes, are colonized by many microorganisms collectively known as a microbiome [24]. This population of microorganisms has a direct impact on both health and the disease. In particular, conclusions have been reached in recent years on how gut microbiota is essential for multiple functions of the host, such as circadian rhythm, metabolism or immunity [25]. Gut microbiota is very important for brain activity [26], as a direct connection can be established to it, represented by what is known as the gut–brain axis. The importance of the gut–brain axis has been highlighted in recent years with regard to the progression of the disease. This axis is characterized by representing a bidirectional communication system, as intestinal functions are controlled by the autonomic and enteric nervous systems [27]. Thus, alterations in the axis have been related to depression [28], anxiety [29], ischemic stroke [30] and symptoms of autism spectrum disorder [31]. Furthermore, currently there is a clear consensus on the impact that these alterations have on the development of the most prevalent neurodegenerative diseases, such as Parkinson’s disease [31], Alzheimer’s disease [32] or multiple sclerosis [33].

Therefore, the objective of our narrative review was to identify those bacteria that could have connection with the mentioned disease (ALS) and to analyze the benefits produced by administering polyphenols. In this sense, an extensive search on the PubMed and EBSCO database was carried out, selecting the most relevant articles published between 2005 and 2020 on research carried out on cell, animal and human models of the disease, using key words, such as microbiota, polyphenols and amyotrophic lateral sclerosis. Based on the obtained papers, a selection was finally analyzed as indicated in Figure 1.

## 2. Gut Microbiota and ALS

There is a diversity of microbial communities in the whole gastrointestinal tract that has a fundamental role in human health. Gut microbiota is made up of microorganisms, mainly bacteria, archaea, fungi and viruses, which are found colonized in the intestine of the human body. Their main role is to maintain homeostasis and its influence on systemic immunity [35]. Bacteria make up 95% of living cellular organisms in the gut microbiota and all of them have different functional capacities and require different nutrients to survive [36]. There are up to 55 phyla of bacteria and 18 of archaea, although only 8 phyla of bacteria have been identified, of which 90% are Firmicutes and Bacteroidetes, while 10% are made up of Actinobacteria and, to a lesser extent, Proteobacteria, Verrucomicrobia and Cyanobacteri [37]. Due to new technologies, the Human Microbiome Project has been developed, which has added 900 sequences that are a reference in the bacterial genome [38]. Nonetheless, the human microbiome composition varies in each individual depending on eating habits, age, coevolution of intestinal microbes with the host, the environment and the microbial ecosystem including three main domains: bacteria, archaea and eukaryotes [37]. Furthermore, microorganisms that live in the gastrointestinal tract synthesize and modify the metabolites produced by the host, and create new metabolites from compounds in the diet, some of which are beneficial and others harmful to a person’s health [39]. In this sense, microorganisms produce short-chain fatty acids (SCFA) and branched chain fatty acids (BCFAs) throughout the fermentation process. Some functions of these fatty acids are, on the one hand, to intervene when differentiating and dividing enterocytes and, on the other hand, maintain and regulate mineral balance, favoring the absorption of magnesium, calcium and iron [40]. Due to all these functions, the role of gut microbiota in physiology has been given great importance, therefore it can be considered as another human organ [41]. It was originally established that the brain was only regulated by the central nervous system (CNS) [42], yet, as previously mentioned, there is recent scientific evidence of bidirectional communication between the intestine and the brain through gut microbiota, therefore microbiota influences the brain’s physiology. In this sense, alterations of the microbiota have been described in patients with Parkinson’s disease, Alzheimer’s disease or multiple sclerosis (MS) [43]. These alterations are related both to aetiology and pathogenesis [44,45]. Several preclinical and clinical studies have indicated that gut microbiota, due to the fact that it maintains physiological homeostasis, can significantly interfere with brain functions and cognitive systems in human beings [46].

Dysbiosis is a state when the gut microbiota is abnormal, [47] where microbial diversity is reduced and there are pathogenic strains or a loss of beneficial strains [48], leading to an intestinal imbalance in the properties and/or functions of gut microbiota. This is due to being exposed to different factors, such as stress, eating habits, drugs (mainly antibiotics) and some toxins and pathogens. These changes create a predisposition in the host to develop certain medical conditions, such as allergies, obesity, intestinal inflammation, diabetes, cancer, atherosclerosis and neurodegenerative diseases. There are two pathways involved in communication between gut microbiota and the CNS: the vagus nerve and the transmission of molecules through the vascular system and the blood–brain barrier (BBB) [49]. Healthy elderly people do not show changes in the diversity of gut microbiota. Studies conducted on centenarians have outlined the importance of microbial diversity to have good health when older [50]. On the other hand, dysbiosis has been observed in neurodevelopmental disorders, for instance autism, and neurodegenerative diseases, such as Alzheimer’s disease, Huntington disease and Parkinson’s disease [51].

In terms of ALS, there is evidence that intestinal dysbiosis is present, and an increase in intestinal permeability in SOD1 (G93A) rodents [52]. This permeability is influenced by low levels of bacterial products as butyrate and SCFA, indeed, increasing these fatty acids results in intestinal integrity improvement, microbial homeostasis and extended life expectancy [53]. The increase in permeability in the intestinal epithelium in ALS is related to dysbiosis, with its consequential systemic dissemination, causing an immune system response that stimulates the action of macrophages and dendritic cells in the production of proinflammatory cytokines, which in turn activate adaptive immune cells that alter immune homeostasis [36]. In terms of bacteria, in the first stage of the disease it has been shown how dysbiosis is characterized by a significant reduction of *Butyrivibrio fibrisolvens* and Firmicutes (they produce butyrate), and a decrease in the expression of the protein involved in tight and adherents junctions, with the consequential increase in the permeability of the intestinal epithelium. We must also add an increase in the number of abnormal Paneth cells that are specialized intestinal epithelial cells whose main functions are to detect microbes and secrete antimicrobial peptides, therefore directly influencing the intestinal environment [54]. A significant increase has also been observed in the *Dorea* bacterial genus that is characterized by having harmful microorganisms, and in *Anaerostipes*; and a significant decrease in the *Prevotella* genus and in some genera of the Lachnospiraceae family, which, on the contrary, are characterized by being comprised of microorganisms that are beneficial for ALS patients. It is important to outline that a decrease in the relation of the Firmicutes/Bacteroidetes phyla has been observed, causing an increase in Bacteroidetes phylum in ALS [55], as also described in the study conducted by Rowin et al. (2017) [56], and associated with a poor quality gut microbiota in babies and the elderly [57]. Furthermore, there is evidence about how there are clear differences between healthy people and patients with the disease, mainly characterized by a low abundance of yeasts and a high abundance of *Escherichia coli* and Enterobacteriaceae in ALS [58]. The study conducted by Zhai et al. (2019) proves how beneficial microorganisms from the *Faecalibacterium* and *Bacteroides* genera are reduced in ALS patients [59]. This dysbiosis is already observed in SOD1 (SOD1-Tg) transgenic mice prone to ALS in a presymptomatic stage, while the exacerbation of symptoms has been associated with the presence of *Ruminococcus torques* bacteria that is linked to neurodegeneration. On the contrary, the presence of *Akkermansia muciniphila* is related to a symptomatic improvement and a possible increase in motor neurons and survival as a result of a rise in nicotinamide [60], which is also associated with motor and functional improvements in ALS patients [61].

Regarding the direct relation of bacteria with pathogenic mechanisms of the disease, again *Akkermansia muciniphila* is outlined due to its capacity to increase GABA/glutamate ratios in the hippocampus [62]. In addition, evidence shows that some species of *Lactobacillus* and *Bifidobacterium* segregate GABA, being the GABA system one of the main mechanisms of brain chemistry modulation by intestinal bacteria [63]. In particular, the *Lactobacillus* species: *L. rhamnosus* and *L. brevis*, and *Bifidobacterium dentium*, increase the GABA neurotransmitter [64,65,66]. On the other hand, gut microbiota interferes in the production of neuroactive metabolites, specifically neurotransmitters and neuropeptides, thus influencing brain functions. In this sense, *Bifidobacterium infantis* bacteria increase the levels of tryptophan in plasma, therefore affecting the central transmission of serotonin [67], and increasing the synthesis of SCFAs that also have neuroactive properties as a result of bacterial fermentation [63].

All of this evidence displays a clear relation between bacterial composition and the development of neurodegeneration, regardless of the region of the brain or type of neuron affected. In this sense, it was maybe Fang P. in 2016 who reviewed this problem in depth for the first time [68]. This was followed by work in the same line with reviews from Spielman LJ et al., 2018; Castillo-Álvarez F et al., 2019 or Roy Sarkar et al., 2019 [49,69,70]. In relation to ALS, the review by Wright et al. in 2019 pointed out that dysbiosis was the etiological cause of the disease, based mainly on the absence of bacteria that produce butyrate in studies with animal models [71]. The review by McCombe PA et al., 2019 also indicated dysbiosis as an important pathogenic mechanism in ALS [36], although accepting the limitation of the work due to the lack of published studies. Based on these studies, later reviews indicate the possibility that improving microbiota can be beneficial for ALS patients. Thus, studies that analyze the impact of microbiota transplantation to improve neurological disorders, [72] or the importance of diet factors in neurodegeneration [73] are outlined. It is especially in relation to the latter and once the main bacteria associated with ALS have been identified, we believe that it would be interesting to analyze the impact that antioxidants in the diet could have.

## 3. Diet, Polyphenols and Neuroprotection. The Effect on Gut Microbiota

Gut microbiota is modulated by diet. Therefore beneficial bacteria are increased with nutrients, including wholegrain oats that increase the number of *Bifidobacterium* (in particular *Bifidobacterium adolescentis*), the production of SCFAs and the levels of the neuroactive substance GABA [74]. Several studies conclude that there is a high alteration in gut microbiota with regard to whether an individual follows a diet rich in fiber or rich in food of animal origin. There is evidence showing that when a diet rich in fiber is followed, the amount of *Prevotella* spp. increases. On the other hand, when a diet rich in food of animal origin or saturated fat is followed for a long period of time, *Bacteroides* rise. Following a diet rich in fiber has been observed to produce an increase in colonic fermentation, leading to a reduction in pH from 6.5 to 5.5, thus favoring butyrate being produced by Gram-positive bacteria and decreasing the growth of *Bacteroides* spp. [75]. Additionally, different studies show that diets rich in fiber and polyphenols have an important role as bacteria growth inhibitors of the *Bacteroides*, *Clostridium* and *Staphylococcus* genera, therefore outlining the role of polyphenols in the diet. In fact, recent research has concluded that there is a direct and positive correlation between the rise in *Lactobacillus* and *Prevotella* and the consumption of fruit and vegetables rich in polyphenols [76]. Further evidence also shows how incubating probiotic bacteria (specifically *Bifidobacterium*) with polyphenols produce SCFAs [77].

Gut microbiota variations caused by taking probiotics, prebiotics, antibiotics and other treatment can be useful to treat some medical conditions. Within the Firmicutes phylum, the bacteria that are most used as a probiotic are *Lactobacillus*, *Bifidobacterium* and some yeasts. In general, these bacteria are added to food, such as yogurt, soy yogurt or are directly ingested through food supplements. There are well-established prebiotics, such as galactooligosaccharides, fructooligosaccharides and inulin, but there are also other putative prebiotics, such as some oligosaccharides, polyphenols, resistant starch, algae and seaweeds, and intestinal metabolites of the host, among which SCFAs are found that can be applied with the aim of selectively and/or differentially affecting beneficial bacteria within the gastrointestinal environment [78,79]. As a result, the use of probiotics and prebiotics must be considered as a good strategy to modulate the gut–brain axis through changes in gut microbiota composition. In this sense, apart from the well-established prebiotics or the SCFAs, the role of polyphenols is also outlined, as ingesting them could represent a key factor in preventing neurodegenerative diseases such as ALS [63]. Polyphenols are bioactive chemical substances that are found in food, mainly fruit and vegetables [80]. In terms of classifying phenolic compounds, they are divided into two main groups: flavonoids and non-flavonoids [81]. There are different subclasses in the flavonoid group, the most relevant are: flavonols, flavones, flavanols, flavanones, isoflavones and proanthocyanidins [82,83,84]. Quercetin stands out among flavonols and it is mainly found in vegetables, such as onion, broccoli and spinach, and green tea, wine and some red fruits [85,86,87]. Apigenin and lutein stand out among flavones, mainly found in celery and parsley [88], and baicalin, produced by *Scutellaria baicalensis* [89]. The most relevant compounds of flavanols are catechins, of which epigallocatechin gallate (EGCG) is highlighted. It is found in large quantities of green tea and black tea leaves, and some nuts, apples, grapes, dark chocolate and red fruits [90,91]. In the subgroup of flavanones, naringin and hesperidin mainly stand out, and they can be found in citric food and tomatoes [92,93]. In terms of isoflavones, daidzein, genistein and glycitein are highlighted, mainly found in soya, green peas and pulses food [91,94]. Proanthocyanidins are found in red fruits, black grapes and pomegranate, outlining cyanidin as a compound [80,92]. Non-flavonoid substances are divided into phenolic acids, lignans, stilbenes and curcuminoids, among others [83]. Phenolic acids are divided into benzoic acids, in which gallic acid is found in tea leaves and fruit, such as cranberries, grapes and strawberries; and cinnamic acids, being caffeic acid the most relevant one, which is found in coffee [95]. Lignans contain enterodiol and enterolactone that are mainly present in fruit and vegetables, like pumpkin, and some seeds, such as flax seed, sesame seed and pumpkin seed. Stilbenes, such as resveratrol and pterostilbene, are found in grapes, cranberries and red fruits [96]. In terms of curcuminoids, curcumin stands out originating from rhizome of turmeric [97].

Among the health benefits that polyphenols display, their role in improving brain functions is especially significant by means of their neuroprotective activity [98,99], as they prevent neuroinflammation, maintain brain homeostasis and promote cognitive functions [100,101]. In relation to this neuroprotection, many polyphenols act as prebiotics as the levels of inflammation are reduced and they contribute to maintaining gut health. This action is due to the fact that they are capable of inhibiting the growth of gut pathogens and stimulating the growth of beneficial bacteria [102]. This causes an impact on the improvement of neuroprotection in neurodegenerative diseases. In this neuroprotective activity, microbial metabolites of polyphenols show protective effects against overproduction of nitric oxide, such as inflammatory cytokines (IL-6 and TNF-α) in BV-2 microglia [103]. This fact confirms what is described by Gasperotti et al. (2015). This author explains that after inoculating a group of mice with a combination of 23 primary polyphenol metabolites, 10 reached the brain, among which gallic acid and caffeic acid stood out. Therefore, it seems evident that the brain is a target organ for metabolites resulting from the degradation of polyphenols [104]. In addition, the metabolism of polyphenols depends on the initial levels of certain bacterial populations. In particular, consuming sour cherries rich in anthocyanins and flavonoids leads to opposite responses depending on the initial levels of *Bacteroides*. Therefore, individuals with a high content of *Bacteroides* respond with a decrease in *Bacteroides* and *Bifidobacterium*, and an increase in Lachnospiraceae, *Ruminococcus* and *Collinsella*. On the other hand, individuals with low levels of *Bacteroides* respond with an increase in *Bacteroides*, *Prevotella* and *Bifidobacterium*, and a decrease in Lachnospiraceae, *Ruminococcus* and *Collinsella* [105]. Ingesting polyphenols modifies microbiota, but microbiota also enhances the effects of polyphenols and modifies them by producing metabolites that improve the prognosis of neurodegenerative diseases. In this sense, different authors have studied said beneficial effects. This is the case of the study by Ho et al., who stated that a rich preparation in flavanols with an α-synuclein reducing activity (which is the key neuropathological hallmark) improved the generation and bioavailability of microbial metabolites of phenolic acid derived from flavanols that have bioactivity to interfere with the incorrect folding or inflammation of α-synuclein [106]. Another example is found with Wang et al. who demonstrated that after administering polyphenolic extracts of grape seeds in mice, they were able to observe how microbiota modified the production of phenolic acids and improved their bioavailability in the brain. Two of these phenolic acids (3-hydroxybenzoic acid and 3-(3-hydroxyphenyl) propanoic acid) strongly interfered with the assembly of β-amyloid peptides and, thus, neurotoxic β-amyloid aggregates [107]. Finally, Sun et al., in the study on APP/PS1 mice (Alzheimer’s disease models), observed that administering curcumin tended to improve spatial learning and memory abilities, and reducing the amyloid plaque load in the hippocampus. These improvements were attributed to changes in the microbiota (an alteration of the relative abundance of bacterial taxa, such as Bacteroidaceae, Prevotellaceae, Lactobacillaceae and Rikenellaceae families, and *Prevotella*, *Bacteroides* and *Parabacteroides* genera), but also metabolites derived from curcumin with neuroprotective effects for the disease [108]. All these studies demonstrate the interaction between microbiota and polyphenols, generating a beneficial tandem in different neurodegenerative diseases. This is why we believe it is important to study this relationship in ALS patients, after treatments with polyphenols that benefit the microbiome and the production of neuroprotective metabolites in the pathology.

## 4. Polyphenols, Microbiota and ALS

In terms of the neuroprotective effect of ALS, after an individual analysis of the polyphenols highlighted in the previous classification was carried out, not all of them have shown improvements in the disease in cellular, animal or human models. However, many of them could offer interesting therapeutic options to treat neurodegeneration of the disease. In this study, the beneficial effects of the main polyphenols in gut microbiota and the possible effects in ALS are shown.

### 4.1. Flavonoids

#### 4.1.1. Quercetin

Quercetin has been observed to influence bacterial flora as it inhibits the growth of *Enterococcus* [109] and increases *Akkermansia muciniphila* [110], which is a bacterium that has been positively correlated to improvements in ALS [60]. Quercetin supplementation has also shown to boost populations of *Bacteroides*, *Bifidobacterium* and *Lactobacillus* that are related to GABA synthesis and that are found to be lower in ALS patients [111]. In terms of its metabolism, quercetin is a polyphenol that needs to be metabolized to be absorbed. The *Eubacterium ramulus* and *Clostridium orbiscindens* bacteria have been shown to intervene in the degradation of this substance, especially in the large intestine. Concentrations of Fusiobacteriaceae and Enterobacteriaceae are directly related to quercetin, inhibiting its degradation by other bacteria. In addition, the amount of Sutterellaceae and Oscillospiraceae is negatively related to the concentration of quercetin [112]. These bacteria have not been related to improvements in ALS patients in previous studies, despite improvements evidenced by the use of this polyphenol. There are mutated forms of the SOD1 protein in familial ALS, they do not fold correctly and are aggregated in motor neurons. Several authors have researched the role of some polyphenols regarding the production of these aggregates. Philbert et al. could verify in vitro how quercetin, quercetin-3-β-d-glucoside (Q3BDG) and, to a lesser extent, EGCG, decreased incorrect folding and, therefore, aggregation induced by hydrogen peroxide in mutated SOD1 protein (A4V SOD1), by means of a direct quantifiable interaction between the three polyphenols and A4V SOD1 [113]. This gives rise to a decrease in toxicity mediated by SOD1 fibrils, showing the great potential that this flavonoid has against A4V SOD1 mutant fibrillation [114]. Finally, quercetin has shown to decrease neuronal damage induced by aluminum, a metal that is precisely linked to the pathogenesis of several neurodegenerative diseases [115].

#### 4.1.2. Catechins

Green tea polyphenols have been shown to have benefits for human health as a result of interacting with gut microbiota. The polyphenols found in tea are epicatechin gallate, EGCG, epigallocatechin, catechin, epicatechin and gallocatechin that inhibit the growth of pathogenic bacteria, such as *Listeria monocytogenes*, *Helicobacter pylori*, *Salmonella typhimurium*, fungi from the genus *Candida, Staphylococcus aureus, Escherichia coli* and *Pseudomonas aeruginosa* [116]. Among these, evidence obtained with EGCG stands out, which has been related to modifications in the composition of gut microbiota, by promoting a significant increase in Verrucomicrobia and Actinobacteria, and a significant reduction in Deferribacteres and Proteobacteria [117]. In addition, its combinations with other polyphenols manage to change bacterial flora. In the study carried out by Most, et al. (2017), it was concluded that supplementation for 12 weeks with the combination of EGCG and resveratrol caused alterations in gut microbiota in obese men, leading to a decrease in Bacteroidetes and *Faecalibacterium prausnitzii* [118]. Furthermore, its synergistic effect with probiotic bacteria stands out. Banerjee, et al. (2019) concluded, after analyzing encapsulation of probiotic bacteria alongside a formula rich in polyphenols characteristic of black tea, white tea, green tea and Thai herbal extract, how this formulation causes an increase in the probiotic effect of these bacteria [77]. Regarding the relation to the disease, EGCG manages to extend life expectancy in ALS animal models [119] due to the protective effect of motor neurons as a result of regulating glutamate levels [120]. This activity seems to be due to inhibiting β-sheet folding that leads to mutations in the SOD1 protein [121], or the increase or improvement of the brain-derived neurotrophic factor (BDNF) obtained in animal models after administering EGCG [122]. These benefits could also be linked to the role of polyphenols in green tea in gut microbiota. In this sense, there has been evidence on how increasing the population of *Oscillospira* [123] could promote muscular improvement in these patients, due to the positive relation of these bacteria with an increase in lean mass [124,125] that has been deteriorated throughout the progression of the disease. Additionally and importantly, there has been recent evidence that administering green tea significantly increases the growth of *Akkermansia muciniphila* [126]. Thus, it is interesting how *Akkermansia muciniphila* is positively associated with Verrucomicrobia, hence promoting its increase can also raise that of Verrucomicrobia, which is also correlated with lower inflammation as serum and tissue levels of inflammatory cytokines and chemokines, such as TNF-α, IL-1α, IL-6 or IL12A, are reduced [127]. In terms of Actinobacteria, their increase is associated with improvements in depression in an inflammation model of depression. The majority of patients with ALS suffer from depression [128], which also significantly influences the progression of the disease [129].

#### 4.1.3. Naringin and Hesperidin

Naringin and hesperidin could also be therapeutic alternatives to reduce inflammation and oxidation in neurodegenerative diseases. These antioxidants are mainly found in oranges and have shown positives changes in gut microbiota. In the study conducted by Fidélix et al. (2020), an increase in *Lactobacillus* spp., *Bifidobacterium* spp. and *Akkermansia* spp. was obtained after administering orange juice [130]. On the other hand, hesperidin raises the proportion of *Lactobacillus/Enterococcus* [131] and *Akkermansia muciniphila* [132]. The metabolism of both polyphenols in the intestine depends on the level of conjugation of sugar fractions that are eliminated by bacteria in the intestine. Naringin and hesperidin practically reach the colon intact, after the enzymes secreted by gut microbiota act (α-rhamnosidase and β-glucosidase), resulting in the formation of their aglycones, hesperetin and naringenin [133]. Regarding the direct impact of neurodegeneration, these compounds have been shown to be able to cross the blood–brain barrier, so they can have neuroprotective and neuromodulating actions in different pathologies of the brain [134]. The neuroprotective effects of naringin have been evidenced in the study by Gopinath et al., where a modulation in the expression of metalloproteases was verified after motor alteration caused by 3-nitropropionic acid. Naringin has been shown to have an anti-inflammatory effect on the brain by decreasing the expression of nuclear factor-kappa B and the glial fibrillary acidic protein [135]. This activity could explain the great benefits observed in diseases such as Alzheimer’s disease and Parkinson’s disease, among others [136]. As regards the neuroprotective potential in ALS, naringin shows positive effects in terms of the aggregation of mutant SOD1 [137,138]. The group of Srinivasan described with quantum chemistry and molecular mechanic calculations how naringin had a strong affinity for mutant SOD1, thereby preventing the formation of toxic aggregates [137]. Joining naringin to the mutated protein occurred due to the OH groups it presents, causing an increase in electrostatic attraction with residual fragments of the mutated protein. In this sense, Zhuang et al. confirmed by means of molecular simulation that naringin binds to SOD, being able to stabilize the SOD1 dimer and inhibit the aggregation of this protein [138]. Hesperidin also displays positive results when inhibiting mutant SOD1 [139]. As previously indicated, hesperidin increases the proliferation of some species of *Lactobacillus* and *Bifidobacterium*. This is associated with a lower loss of motor neurons due to a rise in the secretion of GABA [131]. The proliferation of *Bifidobacterium* could be beneficial for the disease as it promotes the secretion of serum tryptophan [140] and, therefore, of serotonin, which, alongside the production of SCFA through fermentation with *Bifidobacterium*, generate a neuroprotective effect [63].

#### 4.1.4. Genistein

In terms of isoflavones, there is evidence in animal models that genistein has neuroprotective properties for ALS, as it eliminates the production of proinflammatory cytokines and improves gliosis in the spinal cord. Additionally, it improves the viability of motor neurons in animal models, therefore, the effects are expected to be equally beneficial in human ALS models [141]. Regarding its role in gut microbiota, this seems beneficial. It has been observed how an increase in several bacterial strains of *Lactobacillus* spp. is achieved, which are also associated with a 30% rise in SCFA [142]. In addition, the production of microorganisms of the *Clostridium coccoides–Eubacterium rectale* group, *Lactobacillus–Enterococcus* group, *Faecalibacterium prausnitzii* subgroup and *Bifidobacterium* genus was stimulated in postmenopausal women [143,144]. The latter has also been seen to increase in the study conducted by Nakatsu et al., (2014) [145]. An increase in the proportion of Firmicutes/Bacteroidetes seems to be produced in obese men [146].

#### 4.1.5. Proanthocyanidins

*Akkermansia muciniphila* is especially relevant due to the association with specific benefits in ALS, as they are part of the beneficial intestinal flora. Proanthocyanidins favor growth and this process is related to a decrease in obesity and metabolic syndrome [147]. In addition, consuming anthocyanins promotes a significant rise in Actinomycetales, Bifidobacteriacae, Coriobacteriaceae and the proliferation in *Bifidobacterium* spp., which is highly efficient as a probiotic; while inhibiting the growth of *Bacteroides* and *Clostridium histolyticum* that are shown to be pathogenic for humans [40,148]. The increase in *Bifidobacterium* and *Lactobacillus* seems interesting [149] as it can be related to benefits in ALS patients due to the fact that high levels of SCFA are generated with a neuroprotective effect [77]. In this sense, diets enriched with whole cranberry powder and fractions rich in polyphenols not only modify gut microbiota, but also improve fecal SCFA and branched chain fatty acids [150].

#### 4.1.6. Baicalin

Baicalin is the main component of the anti-inflammatory herb *Scutellaria baicalensis* and displays interesting properties to improve gut microbiota, mainly because it increases the population of species of bacteria that produce SCFAs, which was associated with improvements in the abnormal metabolism of glucose and lipids caused by high-fat diets [151]. This beneficial impact could be due to its biotransformation, by the gut microbiota, into baicalein, which has stronger bioactive effects than baicalin itself [152].

All these main results of the interaction between flavonoids and microbiota changes are resumed in the next table (Table 1).

### 4.2. Non-Flavonoids

#### 4.2.1. Gallic Acid

Among non-flavonoids are phenolic acids, where gallic acid stands out due to its great anti-inflammatory activity in the intestine [153]. This molecule has many therapeutic possibilities that still need to be researched. In this sense, we can predict that interesting results in the therapeutic treatment in ALS patients can be achieved, due to the fact that this has been observed in Wistar rats that were induced with neurodegeneration by toxicity caused by aluminum (AlCl_3_), and how they protect motor neurons from the toxicity caused by this metal. This benefit occurs by improving the antioxidant status and by preventing glutamate excitotoxicity [154], which could be mediated by the production of quinolinic acid after degradation of tryptophan through the kynurenine pathway [155]. With relation to microbiota, gallic acid increases probiotic bacteria such as Proteobacteria and Prevotellaceae, and decreases some pathogenic species, mainly in the phyla Firmicutes and Proteobacteria [153]. Regarding metabolites derived from microbiota activity, gallic acid has been observed to increase the production of anti-inflammatory metabolites, such as SCFA in the colon [156].

#### 4.2.2. Caffeic Acid

Caffeic acid has been identified as an antioxidant molecule able to rescue motor neuron-like cells (NSC-34) that express mutated SOD1 associated with ALS [157]. This makes it one of the best polyphenols to fight against the disease. In fact, caffeic acid phenethyl ester has been seen to slow down the disease, by mitigating neuroinflammation and motor neuron death associated with the clinical pathology of ALS in SOD1 mice (G93A) [158]. In addition, it particularly acts as a neuroprotector to stop the activity of glutamate as it suppresses the accumulation of endogenous ROS and restores the potentiality of the mitochondrial membrane, activating the antioxidant enzyme system by increasing the levels of SOD activity and modulating the rise in intracellular activity [159]. In terms of whether these benefits to treat the disease could be linked to changes in microbiota, it is possible that they can, as caffeic acid included in the diet drastically increases the levels of *Akkermansia* bacterium in mice with colitis. This is associated with a decrease in inflammation mediated by the NF-κB pathway [160]. Furthermore, caffeic acid seems to inhibit the rise in Ruminococcaceae, where the *Ruminococcus* genus is found to be related to ALS (in particular *Ruminococcus torques*) [161].

#### 4.2.3. Resveratrol

Within stilbenes, resveratrol has a wide spectrum of therapeutic effects for health despite its limited bioavailability. Its antioxidant and anti-inflammatory effects stand out as they modulate the levels of proinflammatory cytokines (IL-6, IL-16, IL-1β and TNF-α), which gives it great properties as an antiaging and neuroprotective molecule [100]. In terms of its impact and relation to microbiota, there is evidence on how mice treated with this polyphenol show a considerable alteration in the composition of microbiota, characterized by enriched *Bacteroides*, groups Lachnospiraceae NK4A136, *Blautia*, *Lachnoclostridium*, *Parabacteroides* and *Ruminiclostridium* 9, collectively known as RSV microbiota [162], and *Butyrivibrio fibrisolvens* [163] being lower in ALS patients. As is the case with other polyphenols, gut microflora contributes to the metabolism of resveratrol. Bacteria are capable of converting this molecule into dihydro-resveratrol by means of different reactions. A part of dihydro-resveratrol is absorbed and the other is metabolized in conjugated forms (monosulfate and monoglucuronide) that can be easily eliminated in the urine. *Slackia equolifaciens* and *Adlercreutzia equolifaciens* are the main intestinal bacteria involved in reducing resveratrol [164]. These processes related to microbiota can be linked to protecting the body against neurodegeneration, as some bacteria increased due to resveratrol are lower in ALS, as is the case of *Akkermansia muciniphila*. A diet with a strict calorie intake and especially supplemented with the antioxidant resveratrol has been seen to increase the levels of *Akkermansia muciniphila* [165]. It is precisely administering resveratrol that attenuates dysbiosis through rises in these bacteria [166]. These results support those obtained in cell and animal models of the disease. On a cell level, resveratrol, alongside caffeic acid, phenethyl ester and aescultein, are what show the best performance when rescuing primary motor neuron cultures after trophic factor withdrawal [157]. This protective role is confirmed in vitro, as an increase in cell viability is caused by activating sirtuin 1 [167,168]. In fact, this viability compared to that which riluzole obtains is much higher, as the viability of the drug is practically non-existent [169]. These findings were confirmed in SOD1 (G93A) animal models. Conservation and survival of the spinal motor neuron function was observed, which is associated with a higher expression and activation of sirtuin 1 and AMPK, which means there is an energy improvement in the mitochondria [170]. All of this evidence shows that it is possibly one of the most promising polyphenols as an alternative treatment for patients with ALS, even though bioavailability should be improved, perhaps through nanoencapsulation technology with liposomes.

#### 4.2.4. Pterostilbene

An important stilbene is pterostilbene due to its important therapeutic properties. This antioxidant has already been related to ALS improvements due to the effects in animal models where an increase in functionality and survival of motor neurons has been observed [171]. In relation to bacterial microbiota, pterostilbene intake produces a very healthy microbial profile, characterized by an increase in the Verrucomicrobia phylum. However, there is more, as an increase in the *Akkermansia* and *Odoribacter* genera has also been observed [172]. Additionally, changes in microbiota also have an impact on intestinal metabolism that would lead to neuronal improvements. In this sense, it has been shown how pterostilbene has an activity on monoamine oxidases (MAO), which is responsible for the local regulation of serotonin. This neurotransmitter is deficient in ALS and, when pterostilbene is in contact with the gastrointestinal tract, MAOB is inhibited, leading to an increase in serotonin [173].

#### 4.2.5. Curcumin

Curcumin belongs to the group of curcuminoids and is a natural phenol derived from *Curcuma longa*, which has received a lot of interest from the scientific community due to its potential medicinal effects. These effects include anti-inflammatory and neuroprotective activity by directly or indirectly eliminating free radicals. In particular, curcumin significantly eliminates the activity of superoxide dismutase (SOD), which is completed by its ability to stimulate innate immune cells that avoid SOD1 protein folding [174]. Curcumin has also shown to have an indirect antioxidant action to increase plasma catalase activity [175]. Therefore, this means it is a good candidate to treat neurodegenerative diseases [38]. It could be especially efficient to treat ALS as there is evidence that curcuminoids attenuate the expression of COX-2 induced by ROS in ALS. Furthermore, they improve the symptoms of neurological damage caused by heavy metal intoxication linked to ALS [176]. This has an impact on the progression of the disease. We observed that daily intake of 600 mg curcumin reduces the progression of the disease, in parallel with metabolic improvement and a decrease in oxidative damage [177]. In order to obtain bioactive products of curcumin, biotransformation by the human intestinal microflora is necessary. Bidirectionally, curcumin has been shown to have beneficial effects on gut microbiota by increasing the number of bacterial families, including Prevotellaceae, *Bifidobacterium*, Lactobacilli, Bacteroidaceae and Rikenellaceae; and reducing the number of proinflammatory bacterial families, such as Enterobacteriaceae and *Enterococcus* [178,179]. In this sense, there is evidence on how curcumin derivatives (an aggregate with silver), which gives it greater photostability, inhibited the growth of *Escherichia coli* in vitro [180]. When analyzing the impact of curcumin by means of these microbial changes, we can observe that it increases *Prevotellaceae* and *Bifidobacterium*, decreased in the most prevalent neurodegenerative diseases such as Parkinson’s disease, and yet it decreases enterobacteria [181] that are higher in ALS (in particular *Escherichia* and *Enterobacter*) [58]. These changes could mean there is a higher secretion of GABA [63] and tryptophan [140] that, as already indicated, have neuroprotective effects in ALS patients [77]. Furthermore, after ingesting curcumin, an increase in the *Bacteroides* genus is also observed [182], which is lower in ALS patients [58,59]. After the analysis, administration benefits are the most evident and we believe that it should be considered as one of the best alternatives to treat the disease. This is why we consider that new studies should be conducted with ALS patients to objectively assess the scope of these clinical improvements.

All these main results of the interaction between non-flavonoids and microbiota changes are resumed in the next table (Table 2).

## 5. Conclusions

There is currently a general consensus on the benefits and therapeutic potential of polyphenols when treating neurodegenerative diseases. Many of the improvements occur through the activity of these antioxidants on gut microbiota, by decreasing the bacteria that are pathogenic and increasing those that are beneficial. Consequently, they generate metabolites and neurotransmitters that protect neuronal activity. In the specific case of ALS, Figure 2 shows all these changes, analyzed throughout our work. Based on these alterations, all the studies reviewed in this article related to ALS conclude that polyphenols are a therapeutic alternative to improve prognosis. However, quercetin naringin, hesperidin, genistein, cyanidin and gallic acid have not been directly related to improvements in the disease, although they do achieve effects in the microbiota, which are promising.

In terms of antioxidants that have been used successfully in ALS, it is worth highlighting EGCG that achieves protective motor neuron effects associated with regulating the level of glutamate, thus avoiding misfolding of the SOD1 protein. In addition, it is interesting to review the benefits obtained when combined with resveratrol or probiotic bacteria, as they have also been associated with increases of *Akkermansia muciniphila*. Resveratrol precisely manages to improve the disease at the cellular, animal and human level, which is completed with a very beneficial impact on gut microbiota, enriching it with anti-inflammatory bacteria, among which *Akkermansia muciniphila* stands out. In relation to curcumin there have been very good results in ALS patients in reducing high oxidative stress, acting in various molecular mechanisms involved in excess oxidation. In addition, it improves gut microbiota and, in particular, bacteria that have been altered in ALS, such as *Escherichia*, *Enterobacter*, *Clostridium* and *Bacteroides*.

Based on these conclusions, Figure 2 summarizes the benefits of the main polyphenols in the pathogenesis of the disease. Specifically, we highlighted the benefits of resveratrol, EGCG and curcumin, which are shown to be particularly relevant in the studies analyzed.

Despite these findings, studies related to the benefits of polyphenols in gut microbiota of ALS patients are still at a very early stage. It is necessary to delve into the results analyzed in our review in order to establish all molecular mechanisms that antioxidants may use, which could clarify our knowledge on the pathogenesis of the disease and augment current treatments.

## Figures and Tables

**Figure 1 metabolites-11-00120-f001:**
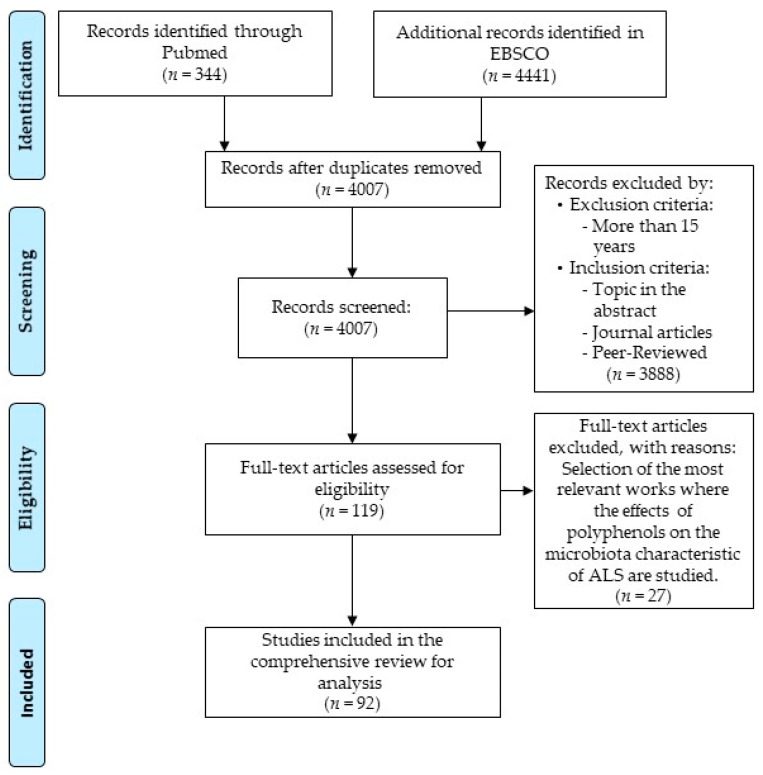
Consort Flow, based on the Preferred Reporting Items for Systematic Reviews and Meta-Analyses (PRISMA) template [34].

**Figure 2 metabolites-11-00120-f002:**
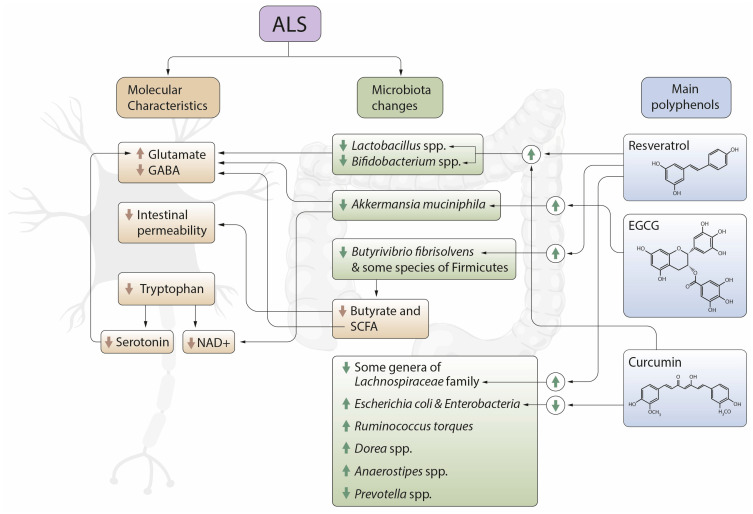
Relationship between microbiota and molecular changes in amyotrophic lateral sclerosis (ALS) patients and impact of the polyphenols resveratrol, epigallocatechin gallate (EGCG), resveratrol and curcumin on the levels of some of the bacteria most directly related to the disease, as well as molecular changes at the neuronal level also linked to the pathogenesis of ALS. GABA: γ-aminobutyric acid; SCFA: Short chain fatty acids; NAD+: Nicotinamide riboside.

**Table 1 metabolites-11-00120-t001:** Flavonoids effects on intestinal microbiota.

Flavonoid	Microbiota Change	Study Population	Author, Year
Quercetin	↓*Enterococcus* spp.	Bacterial culture with Broth medium	Firrman, et al. (2016) [81]
	↑*Akkermansia muciniphila*	Wistar rats	Etxeberria, et al. (2015) [82]
	↑*Bacteroides* spp. ↑*Bifidobacterium* spp. ↑*Lactobacillus* spp.	Female C57BL/6 mice	Lin et al. (2019) [83]
EGCG	↑Verrucomicrobia ↑Actinobacteria	Male C57BL/6N mice	Ushiroda et al. (2019) [89]
	↓Bacteroidetes	Overweight and obese men and women	Most, et al. (2017) [90]
	↑*Akkermansia muciniphila*	Male C57/BL6 mice	Jeong et al. (2020) [98]
Naringin & Hesperidin	↑*Bifidobacterium* spp. ↑*Akkermansia* spp.	Female volunteers	Fidélix et al. (2020) [102]
	↑*Lactobacillus* spp.	Female volunteers	Fidélix et al. (2020) [102]
		Male Lewis rats	Estruel-Amades et al. (2019) [103]
	↓*Enterococcus* spp.	Male Lewis rats	Estruel-Amades et al. (2019) [103]
Genistein	↑*Lactobacillus* spp.	In vitro ecosystem with vessels	De Boever et al. (2000) [114]
	↑*Bifidobacterium* spp.	Postmenopausal Caucasian women	Bolca et al. (2007) [115]
		Postmenopausal women	Clavel, et al. (2005) [116]
	↑Firmicutes/Bacteroidetes	Male subjects	Fernández-Raudales et al. (2012) [118]
Proanthocyanins	↑*Bifidobacteriaceae* ↑*Coriobacteriaceae* ↑*Bifidobacterium* spp. ↓*Bacteroides* spp. ↓*Prevotella* spp. ↓*Clostridium histolyticum*	Systematic review of clinical trials	Saez-Lara et al. (2015) [120]

Effects of the main flavonoids on the intestinal microbiota. EGCG: Epigallocatechin-3-gallate.

**Table 2 metabolites-11-00120-t002:** Non-flavonoids effects on intestinal microbiota.

Non-Flavonoid	Microbiota Change	Study Population	Author, Year
Galic acid	↑Proteobacteria ↑*Prevotellaceae*	Male BALB/c mice	Pandurangan et al. (2015) [125]
Caffeic acid	↑*Akkermansia muciniphila*	Female C57BL/6 mice	Zhang et al (2016) [132]
	↓*Ruminococcaceae*	Male Wistar rats	Zhang et al (2017) [133]
Resveratrol	↑*Bacteroides* spp. ↑Lachnospiraceae NK4A136 ↑*Blautia* spp. ↑*Lachnoclostridium* spp. ↑*Parabacteroides* spp. ↑*Ruminiclostridium 9*	C57BL/6J mice	Wang et al. (2020) [134]
	↑*Butyrivibrio fibrisolvens*	Thin-tailed Han cross-bred ewes	Ma et al. (2015) [135]
	↑*Akkermansia muciniphila*	Obese men with metabolic syndrome	Walker et al. (2018) [137]
Pterostilbene	↑Verrucomicrobia ↑*Akkermansia muciniphila* ↑*Odoribacter* spp.	Zucker (fa/fa) rats	Etxeberria et al. (2017) [144]
Curcumin	↑*Bifidobacterium* spp. ↑*Lactobacillus* spp.	C57BL/10ScSn (wildtype) mice	Bereswill et al. (2010) [150]
	↓*Enterobacteria*	C57BL/10ScSn (wildtype) mice	Bereswill et al. (2010) [150]
	↑Bacteroidaceae (*Bacteroides* spp.) ↑Rikenellaceae	C57BL/6 mice	Shen et al. (2017) [151]
	↓*Escherichia coli*	In vitro study with *E. coli*	Abdellah et al. (2018) [152]

Effects of the main non-flavonoids on the intestinal microbiota.

## Data Availability

Not applicable.

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
