# Peer review of "The Impact of Microbiota on the Pathogenesis of Amyotrophic Lateral Sclerosis and the Possible Benefits of Polyphenols. An Overview"

_metabolites, 2021, doi:10.3390/metabo11020120_

Round 1

Reviewer 1 Report

The paper has been improved. Some minor editing of English, where I pointed out in the comments would be desirable.

Reviewer 2 Report

The authors presented a  previous submitted review manuscript where the relationship between gut microbiota, neurodegenerative diseases, and the possible benefits of polyphenols, as a therapeutic option were debated. Overall, I can conclude that the manuscript was improved, the type of review was changed from systematic review to narrative review, that is indeed more adequate for the presented work. As a result of my revision I still have some suggestions and comments. Please find them listed below.

Line 23 – replace “thanks” with “due”

Line 24 – rephrase the part “related to bacteria that could be related to the disease”. The world “related” should appear just one time in the same sentence

Lines 23-26 – please rephrase the whole sentence like this: “Thus, the objective of our narrative review was to identify those bacteria that could have connection with the mentioned disease (ALS) and to analyze the benefits produced by administering polyphenols”

Lines 26-28 – the sentence ““Thus, the objective of our narrative review … administering polyphenols” should be moved in the introduction part.

Lines 29-33 the sentence “Thereby, after selecting…. stand out negatively for the disease.” Is too long. Please split it in two sentences.

Lines 39-40 – please cut “mainly in the levels of the bacteria most positively or negatively related to the disease..” and finish the sentence with “in microbiota” as in the previous version.

The introduction was improved. I have no further comments related to this part.

Figures 1 and 3 are redundant. Please delete Figure 1, keep Figure 3 only, that can be composed from part A (the current Figure 1) and Part B (that part that make the differences between them).

Lines 218-224 – please move the whole paragraph at the end of Introduction, together with Figure 2. By the way, the Figure 2 has low resolution. Please improve it. If is impossible to improve it in the used template, simply draw it power point.

Chapter 2 and 3 were also improved.

Chapter 4 was good from the beginning, however new information was added.

Please put the Tables in the Journal format. This is provided as an example in the original template.

Reviewer 3 Report

Manuscript Metabolites-1124614 entitled "The impact of microbiota on the pathogenesis of amyotrophic lateral sclerosis and the possible benefits of polyphenols. A comprehensive review" by Julia Casani-Cubel et al is a resubmitted paper.

The authors re-submit the manuscript in its revised form, taking into consideration the reviewer comments. The paper seems to have been accurately revised.

All my concerns have been addressed.

Author Response

We thank the reviewer for the comment and express our gratitude for the review.

Reviewer 4 Report

The authors have satisfactorily responded to the comments that the subject matter of this work is acceptable for publication.

Author Response

Thank you very much for your resolution, and we are pleased that you find our work publishable in this journal.

This manuscript is a resubmission of an earlier submission. The following is a list of the peer review reports and author responses from that submission.

Round 1

Reviewer 1 Report

Introduction is ok, classification of the polyphenols, also. Tables are ok. Description of the polyphenolic compounds action, especially on protein folding and aggregation could be improved - where they mention such literature data, let them check original papers, what kind of inhibition of protein aggregation takes place.

Conclusions could be more condensed and conclusive.

Reviewer 2 Report

Major comments

The authors tried to present a review of the relationship between gut microbiota and neurodegenerative diseases and the possible benefits of polyphenols, as a therapeutic option for ALS, through changes induced in microbiota. The topic is interesting and sound, but unfortunately, I cannot advise the publication of the article in the present form. I do consider that the authors should make the effort and re-write the whole article based on my suggestion, and also using some good reviews articles as examples.

Firstly, the present review does not include the mandatory points that should be presented in a comprehensive review. To help the authors I will point below the points that should be tackled:

  • The goal of the review should be presented, the period covered, to which group is addressed (should be presented in the abstract).
  • What was reviewed before and what the present review is adding more; please explain, based on the previous reviews in the field why it is still a room for a new review of the topic (should be presented at the end of the introduction).
  • How the articles search was realized (in which data bases), how the articles selection was preformed and which excluding criteria were applied (this should be a short subheading after introduction). A diagram (for example a consort diagram) can be used to summarize the flow.

The Introduction could be suitable for a research article for example, but not for a review. Please make it more comprehensive. To help the authors in this point I recommend them to read the article published in “Microorganisms” by Mark Obrenovich and co-authors, entitled “The Role of the Microbiota–Gut–Brain Axis and Antibiotics in ALS and Neurodegenerative Diseases”

In the next two parts “Microbiota and ALS” and “Diet, polyphenols and neuroprotection. The effect on gut microbiota” the authors only described some findings. I do consider that in a comprehensive review the findings should be presented in a comparative and critical manner, followed by some discussions where the authors are presenting their own opinions and conclusions.

The last part “Polyphenols, microbiota and ALS” is more related to the review topic, but unfortunately not strong enough. In addition to the described findings, the authors should improve this part also by adding their own discussions.

Minor comments

Line 25 - The construction “most used in other studies, through microbiota improvements” does not make sense. please rephrase it or delete it.

Line 28 – Please replace “On the other hand” with “however”

Line 31 – please replace “that we consider” with “considered”

Please keep a space between the reference number and the last word. For example “which is positively correlated with the severity of the disease [7]”, not “which is positively correlated with the severity of the disease[7]”. Check and correct in the whole text.

Line 64. The sentence “On the other hand, there has been evidence that there is a link between neurotoxicity due to high levels of heavy metals and the disease [16]” does not fit here because your review is not about the effect of heavy metals.

The text between lines 155-280 has a different format. The same in case of the text from lines 307-483.

Please put the Tables in the journal format presented in the template and keep the tables as simple as possible.

Some references are old, but however the authors did not mention the revised period. I suggest them for example to review just the articles form the last 10 years, but of course it is their choice.

Reviewer 3 Report

Manuscript Metabolites-1069082 entitled "The impact of microbiota on the pathogenesis of amyotrophic lateral sclerosis and the possible benefits of polyphenols. A comprehensive review." by Julia Casani-Cubel et al. is a review article and its aim is to summarize the beneficial effect of polyphenols on the microbiota in patients with amyotrophic lateral sclerosis.
I do consider that the article, being a narrative review, is written in an understandable way. The text is clear and easy to read. The results are clearly presented. The authors summarized the data on the topic they proposed. References are up to date.

The article is well written. There are only minor flaws surrounding this work.

1) Abstract: please, remove the last sentence.

2) Table 2: I suggest increasing the resolution.

3) The authors could add a final figure that summarizes the positive connection between microbiota and polyphenols on the pathogenesis of amyotrophic lateral sclerosis.

4) A more correct title is "The impact of microbiota on the pathogenesis of amyotrophic lateral sclerosis and the possible benefits of polyphenols: an overview ".
5) There is a formatting error between lines 281 and 306 (but this often happens during text conversion).

Reviewer 4 Report

This is a well-done review of the impact of microbiota on the pathogenesis of amyotrophic lateral sclerosis and the possible benefits of polyphenols. The literature is summarized well.  

Comments:

  1. In the manuscript, the authors reviewed the diet, polyphenols, neuroprotection and the effect on gut microbiota. The abstract seems like describe the current situation, but not summarize the main points of the manuscript. It should be revised accordingly.
  2. Please provide more detail information about the polyphenols, microbiota and ALS in clinical studies, if there is any available.
